# Isolation of MERS-related coronavirus from lesser bamboo bats that uses DPP4 and infects human-DPP4-transgenic mice

Susanna K. P. Lau [1,2,3,6✉], Rachel Y. Y. Fan[1,6], Longchao Zhu[1,6], Kenneth S. M. Li [1,6], Antonio C. P. Wong [1], Hayes K. H. Luk [1], Emily Y. M. Wong[1], Carol S. F. Lam[1], George C. S. Lo[1], Joshua Fung[1], Zirong He[1], Felix C. H. Fok[1], Rex K. H. Au-Yeung [4], Libiao Zhang[5], Kin-Hang Kok [1], Kwok-Yung Yuen [1,2,3,6] & Patrick C. Y. Woo [1,2,3✉]

While a number of human coronaviruses are believed to be originated from ancestral viruses in bats, it remains unclear if bat coronaviruses are ready to cause direct bat-to-human transmission. Here, we report the isolation of a MERS-related coronavirus, *Tylonycteris*-bat-CoV-HKU4, from lesser bamboo bats. *Tylonycteris*-bat-CoV-HKU4 replicates efficiently in human colorectal adenocarcinoma and hepatocarcinoma cells with cytopathic effects, and can utilize human-dipeptidyl-peptidase-4 and dromedary camel-dipeptidyl-peptidase-4 as the receptors for cell entry. Flow cytometry, co-immunoprecipitation and surface plasmon resonance assays show that *Tylonycteris*-bat-CoV-HKU4-receptor-binding-domain can bind human-dipeptidyl-peptidase-4, dromedary camel-dipeptidyl-peptidase-4, and *Tylonycteris pachypus*-dipeptidyl-peptidase-4. *Tylonycteris*-bat-CoV-HKU4 can infect human-dipeptidyl-peptidase-4-transgenic mice by intranasal inoculation with self-limiting disease. Positive virus and inflammatory changes were detected in lungs and brains of infected mice, associated with suppression of antiviral cytokines and activation of proinflammatory cytokines and chemokines. The results suggest that MERS-related bat coronaviruses may overcome species barrier by utilizing dipeptidyl-peptidase-4 and potentially emerge in humans by direct bat-to-human transmission.

[1] Department of Microbiology, The University of Hong Kong, Hong Kong, China. [2] Carol Yu Centre for Infection, The University of Hong Kong, Hong Kong, China. [3] State Key Laboratory of Emerging Infectious Diseases, The University of Hong Kong, Hong Kong, China. [4] Department of Pathology, The University of Hong Kong, Hong Kong, China. [5] Guangdong Key Laboratory of Animal Conservation and Resource Utilization, Guangdong Public Laboratory of Wild Animal Conservation and Utilization, Guangdong Institute of Applied Biological Resources, Guangzhou, China. [6] These authors contributed equally: Susanna K. P. Lau, Rachel Y. Y. Fan, Longchao Zhu, Kenneth S. M. Li, Kwok-Yung Yuen. ✉email: skplau@hku.hk; pcywoo@hku.hk

Bats are major reservoirs of diverse coronaviruses (CoVs) and believed to be the primary origin of recent human CoV epidemics caused by severe acute respiratory syndrome coronavirus (SARS-CoV) and Middle East respiratory syndrome coronavirus (MERS-CoV)[1–3]. While these flying mammals are likely the ultimate origin of SARS-CoV-2, the immediate source of COVID-19 remains unidentified[4]. MERS-CoV was likely to have originated from bats before it jumped to camels and humans, although existing bat CoVs under the *Betacoronavirus* subgenus *Merbecovirus* (previously named *Betacoronavirus* Lineage C, which contains MERS-CoV) were not close enough to represent the immediate ancestral virus. Yet, some bat merbecoviruses, such as *Tylonycteris*-bat-CoV-HKU4 (*Ty*-BatCoV HKU4), possess spike (S) proteins capable of utilizing the MERS-CoV receptor, human-dipeptidyl-peptidase-4 (hDPP4), in pseudovirus assays[5–12]. However, none of these bat merbecoviruses have been successfully isolated in vitro, which has hampered understanding of their potential for emergence and direct bat-to-human transmission.

In this work, a MERS-related CoV, *Ty*-BatCoV HKU4, is directly isolated from lesser bamboo bats using human colorectal adenocarcinoma (Caco-2) cells. The virus also replicates efficiently in human hepatocarcinoma (Huh7) cells and uses hDPP4 and dromedary camel-DPP4 (dcDPP4) as cell entry receptors. *Ty*-BatCoV HKU4 infects hDPP4 transgenic mice with lung and brain pathologies. The results show that MERS-related bat CoVs may overcome species barrier by utilizing DPP4 from different hosts, with the potential for direct bat-to-human transmission.

## Results

**Isolation of *Ty*-BatCoV HKU4 using Caco-2 cells**. We conducted a 6-year surveillance study of merbecoviruses in 6086 alimentary samples from 51 different bat species from Hong Kong and mainland China (Supplementary Fig. 1a). *Ty*-BatCoV HKU4 was detected in 32 (3.07%) of 1044 samples from lesser bamboo bats (*Tylonycteris pachypus*) in Hong Kong, Guizhou or Guangxi using reverse transcription-PCR (RT-PCR) and sequencing for a partial RNA-dependent RNA polymerase (RdRp) gene fragment and S1 gene (Supplementary Table 1 and Supplementary Fig. 1b). We obtained 14 isolates of *Ty*-BatCoV HKU4 from positive samples using Caco-2 cells, with cytopathic effects (CPE) observed during the first blind passage. Purified viral particles displayed typical CoV morphology of around 100–120 nm in diameter under electron microscopy (Fig. 1e). The full-length genome sequence (30,136 bp) of selected strain SM3A was determined, sharing 97.6% and 68.7% nucleotide identities to the genomes of previously reported *Ty*-BatCoV HKU4 and MERS-CoV strains, respectively (Supplementary Fig. 1c). Upon passages, *Ty*-BatCoV HKU4 SM3A showed one synonymous and one nonsynonymous (G22997T) substitutions at the 5th passage, resulting in Q476H in the S gene; and five synonymous and one nonsynonymous (A12899G) substitutions at the 15th passage, resulting in H4208R in ORF1ab.

**Cellular tropism of *Ty*-BatCoV HKU4**. To determine the potential for interspecies transmission and if *Ty*-BatCoV HKU4 may possess broad tissue tropism similar to that of MERS-CoV, various cell lines were tested for susceptibility to *Ty*-BatCoV HKU4 (Supplementary Table 2). The virus replicated efficiently in Caco-2 and Huh7 cells, with viral loads of $9.6 \times 10^{10}$ copies/ml and $1.3 \times 10^{10}$ copies/ml on day 5, respectively. CPE, mainly consisted of rounding up of fused and granulated cells progressively detaching from the monolayer forming masses of dead cells, could be observed in both cell lines on day 5, which showed viral nucleocapsid expression by immunofluorescence (IF) assay

in 50% of cells (Fig. 1a–d). *Ty*-BatCoV HKU4 was unable to replicate efficiently in other human, bat, or camel cell lines, including normal human bronchial epithelial (NHBE) cells derived from primary airway epithelial cells (Fig. 2a, b).

**Ty-BatCoV HKU4 replication is inhibited by interferons-α/β**. Since type I interferon (IFN) response may serve as a barrier against CoV infections, we also tested the sensitivity of *Ty*-BatCoV HKU4 to IFN response[13]. Pretreatment of Huh7 with IFN-α A/D and IFN-β reduced replication of both MERS-CoV EMC and *Ty*-BatCoV HKU4 SM3A, with more significant reduction of MERS-CoV replication (Supplementary Fig. 2a, b). Notably, pretreatment of Huh7 with both IFNs prevented the formation of CPE in *Ty*-BatCoV HKU4-infected cells (Supplementary Fig. 2c). The results suggested IFNs as potential antiviral candidates against diverse merbecoviruses.

**MERS-CoV neutralizing antibody does not neutralize Ty-BatCoV HKU4**. To assess potential serological cross-neutralization between MERS-CoV and *Ty*-BatCoV HKU4, neutralization assays were performed using 16 serum samples from MERS-infected camels from Dubai with MERS-CoV neutralizing antibody titers of 10–160[14]. None of the samples neutralized *Ty*-BatCoV HKU4, suggesting little cross-antigenicity (Supplementary Table 3).

**Ty-BatCoV HKU4 uses hDPP4 and dcDPP4 as receptors**. To determine if live *Ty*-BatCoV HKU4 can utilize DPP4 as receptor, virus infectivity studies using HEK293T cells expressing or not expressing DPP4 proteins from humans, dromedaries or lesser bamboo bats were performed. *Ty*-BatCoV HKU4 SM3A can use hDPP4 and dcDPP4 as receptors and replicated efficiently in HEK293T cells upon their expression with $>1.5 \times 10^4$ and $1 \times 10^3$-fold increase in viral load, respectively (Fig. 2d). However, it cannot replicate efficiently in *Tylonycteris pachypus* DPP4 (*Tp*DPP4)-expressing HEK293T cells or *Tp*DPP4-expressing primary *Tylonycteris pachypus* kidney and lung cells (Fig. 2c, d). To further confirm if *Ty*-BatCoV HKU4 can use hDPP4, we performed infectivity assays using small interfering RNA (siRNA) knockdown and CRISPR-Cas9 knockout of hDPP4 in Huh7 cells. The infectivities of both *Ty*-BatCoV HKU4 and MERS-CoV were significantly attenuated in both hDPP4-knockdown and hDPP4-knockout cells compared to control Huh7 cells (Fig. 2e, f).

**HKU4-RBD binds to hDPP4, dcDPP4, and TpDPP4**. While previous studies based on sequence and binding analysis showed that *Ty*-BatCoV HKU4-receptor-binding-domain (HKU4-RBD) can bind to hDPP4, its relative binding affinity to dcDPP4 and *Tp*DPP4 was not examined. The S1-RBD sequences of *Ty*-BatCoV HKU4 strains possessed 54.5% aa identities to that of MERS-CoV, with $K_a/K_s$ ratio of 0.067, suggesting purifying selection. Of the 14 critical residues in MERS-RBD important for hDPP4 binding[12,15], five were conserved[16]. Most of the 14 residues in hDPP4 were also conserved in *Tp*DPP4 and dcDPP4, except I → K substitution at position 295 in *Tp*DPP4 and T → V substitution at position 288 in dcDPP4, suggesting possible binding of HKU4-RBD to *Tp*DPP4 and dcDPP4. We examined the binding affinity of HKU4-RBD to hDPP4, dcDPP4 and *Tp*DPP4 in comparison to MERS-RBD using flow cytometry, co-immunoprecipitation (co-IP) and surface plasmon resonance (SPR) assays (Fig. 3). Flow cytometry using HEK293T cells transfected with hDPP4, dcDPP4, and *Tp*DPP4 showed that MERS-RBD bound hDPP4 and dcDPP4 with higher affinity than *Tp*DPP4, whereas HKU4-RBD bound *Tp*DPP4 with slightly higher affinity than hDPP4 and dcDPP4 (Fig. 3a). Co-IP assays

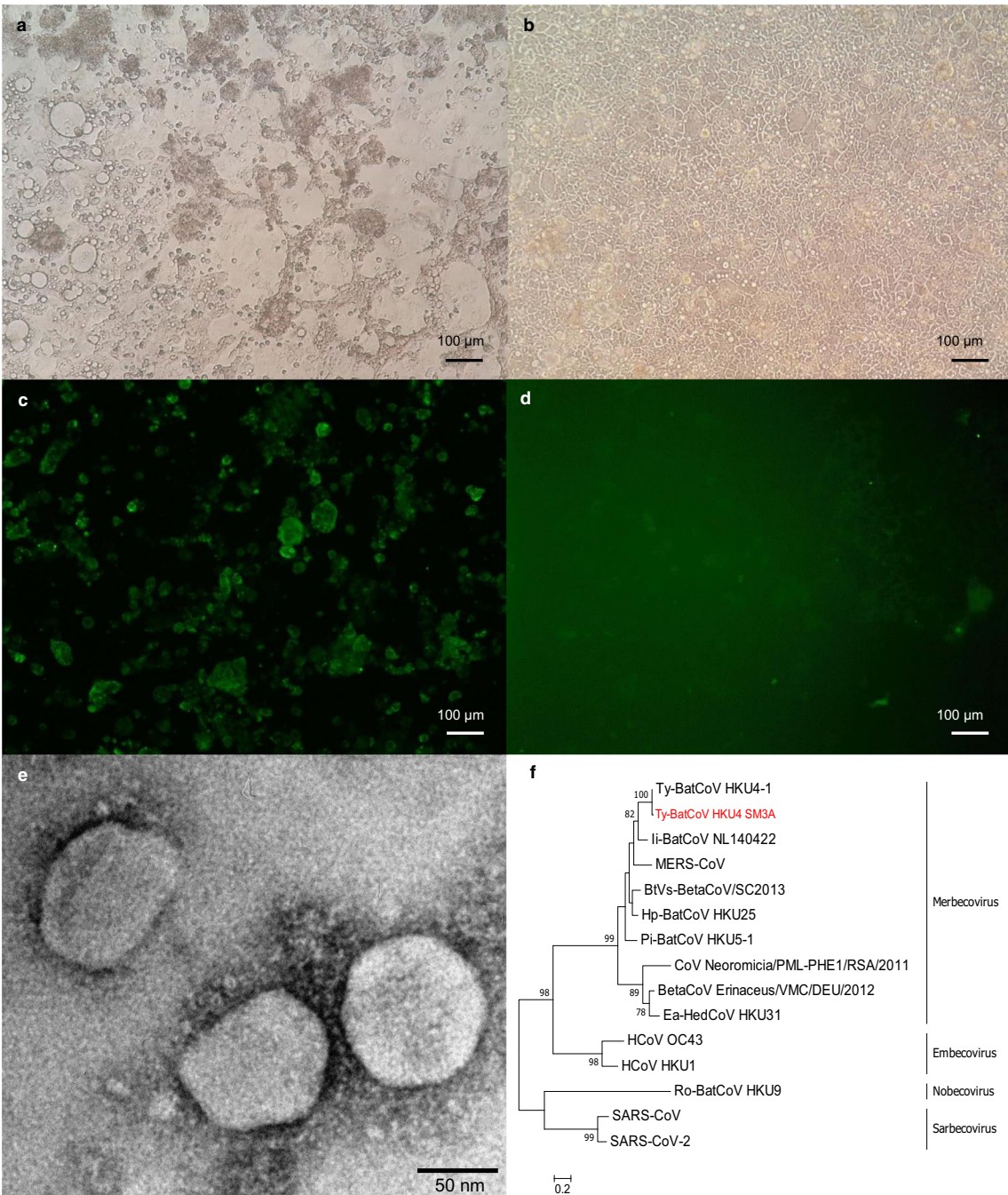

**Fig. 1 Isolation of *Ty*-BatCoV HKU4 using Caco-2 cells. a** Caco-2 cells infected with *Ty*-BatCoV HKU4 SM3A showing CPE with rounding up of fused and granulated cells progressively detaching from the monolayer forming masses of dead cells at 5 dpi, compared to **b** uninfected cells. **c** IF staining of Caco-2 cells infected with *Ty*-BatCoV HKU4 SM3A, compared to **d** uninfected cells using mouse antiserum against N. Scale bars, 100 μm. **e** *Ty*-BatCoV HKU4 SM3A viral particles under transmission electronic scope. Viral particles, size about 100–120 nm, were visualized, showing spikes around the particles typical of coronavirus. Scale bar, 50 nm. Images are representative of three independent experiments. **f** Phylogenetic analysis of S1 amino acid (aa) sequences of *Ty*-BatCoV HKU4 SM3A (red in color) and other selected betacoronaviruses.

showed that MERS-RBD can pull down hDPP4 and dcDPP4 but not *Tp*DPP4, whereas HKU4-RBD can pull down all three DPP4 receptor proteins (Fig. 3b). SPR assays using Biacore X100 showed that MERS-RBD and HKU4-RBD bound to hDPP4 with an equilibrium dissociation constant ($K_d$) of 631.7 nM and 1.103 μM, respectively, suggesting stronger binding between MERS-RBD and hDPP4 by two-fold. MERS-RBD and HKU4-RBD bound to dcDPP4 with $K_d$ of >18.68 and >47.66 μM, respectively, suggesting weaker binding than to hDPP4. MERS-RBD and HKU4-RBD bound to *Tp*DPP4 with $K_d$ of 0.7 M and

2.583 μM, respectively, suggesting much lower binding between MERS-RBD and *Tp*DPP4 (Fig. 3c). The ability of HKU4-RBD to bind hDPP4, dcDPP4, and *Tp*DPP4 albeit with low affinity suggests that *Ty*-BatCoV HKU4 may have the ability to overcome species barrier by using diverse mammalian DPP4 proteins.

**_Ty_-BatCoV HKU4 infects hDPP4 transgenic mice**. We studied the infectivity of *Ty*-BatCoV HKU4 in hDPP4 transgenic C57BL/ 6 N mice by intranasal inoculation (Fig. 4 and Supplementary

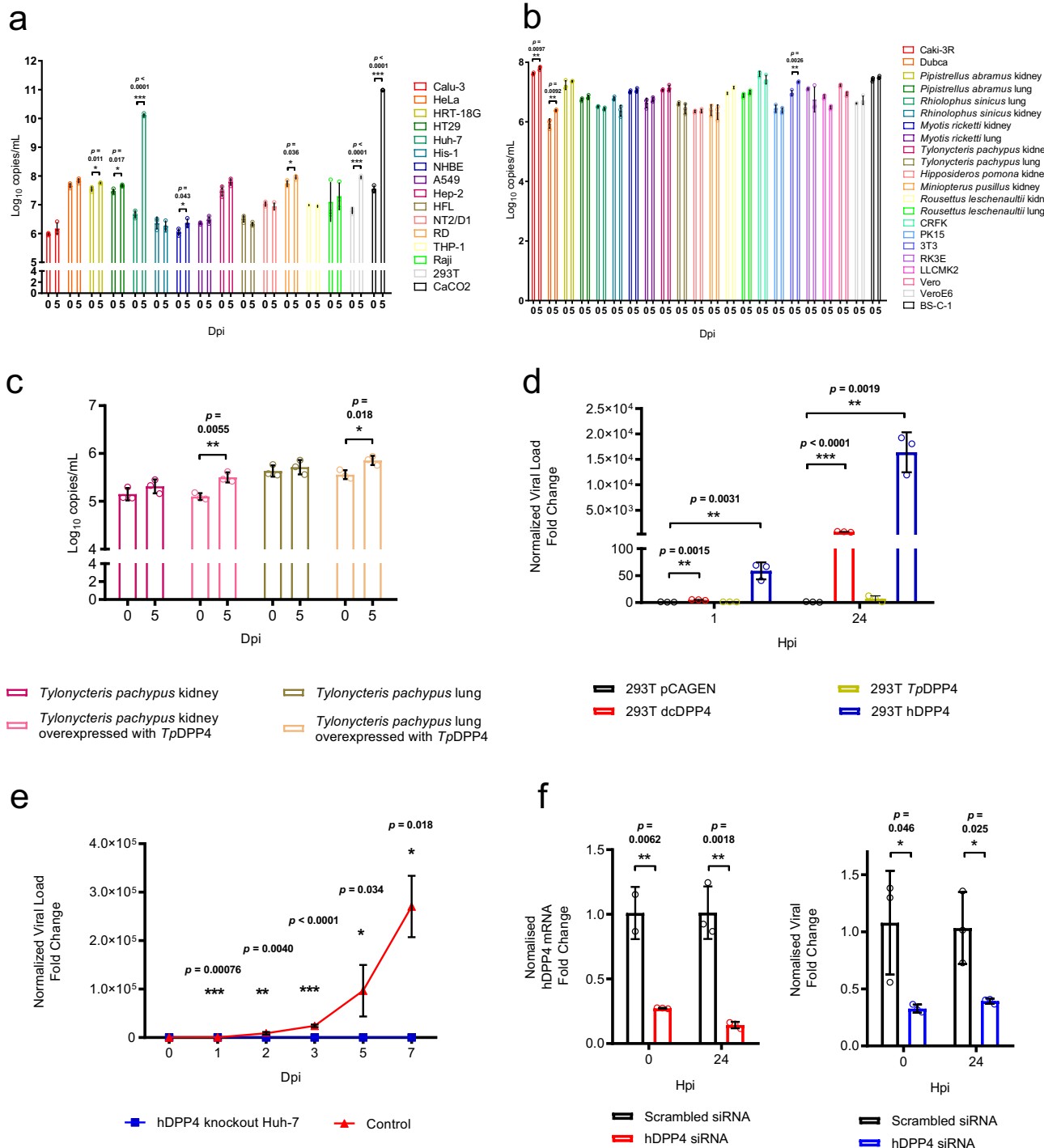

**Fig. 2 Cellular tropism and receptor usage of Ty-BatCoV HKU4 SM3A.** *Ty*-BatCoV HKU4 SM3A of 1 MOI was inoculated to **a** 16 human cell lines; **b** 12 bat and 10 other mammalian cell lines; **c** primary *Tylonycteris pachypus* kidney and lung cells with overexpressed *Tp*DPP4; **d** different host-DPP4-expressing 293 T cells; **e** hDPP4 CRISPR-knockout Huh7 cells; and **f** siRNA hDPP4-knockdown Huh7 cells. Culture supernatants (**a**–**c**, **e**, **f**) and cell lysates (**d**) were harvested from respective timepoints and viral titers were determined by RT-qPCR and normalized to β-actin gene. **f** siRNA efficiency was determined by measuring hDPP4 mRNA expression level in knockdown Huh7 cell lysate at 0 and 24 hpi and compared with mock-treated samples. Data are presented as mean values ± SD, $n = 3$ independent biological replicates for each cell line at each time point. Dots in each graph represent individual samples. The p-values calculated by multiple two-tailed unpaired *t*-test without correction for multiple comparisons (<0.05) are indicated in each graph. Statistical significances are indicated by the asterisks (*$P < 0.05$; **$P < 0.01$; ***$P < 0.0001$).

Table 4). All mice survived till 28 days post-infection (dpi) (Fig. 4a). *Ty*-BatCoV HKU4 was detected in the lungs, spleens and/or brains of infected mice. Reverse transcription quantitative PCR (RT-qPCR) showed the highest viral loads in lung (2.84 $\log_{10}$ copies/mg) and brain (2.99 $\log_{10}$ copies/mg) on 4 and 14 dpi

respectively, with viral clearance on 28 dpi (Fig. 4b). RT-qPCR showed that hDPP4 was stably expressed in different tissues (Supplementary Fig. 3). Histology of lung tissues showed moderate to marked degree of lymphocytic infiltration within the alveolar septa and macrophages in the alveoli, compatible with

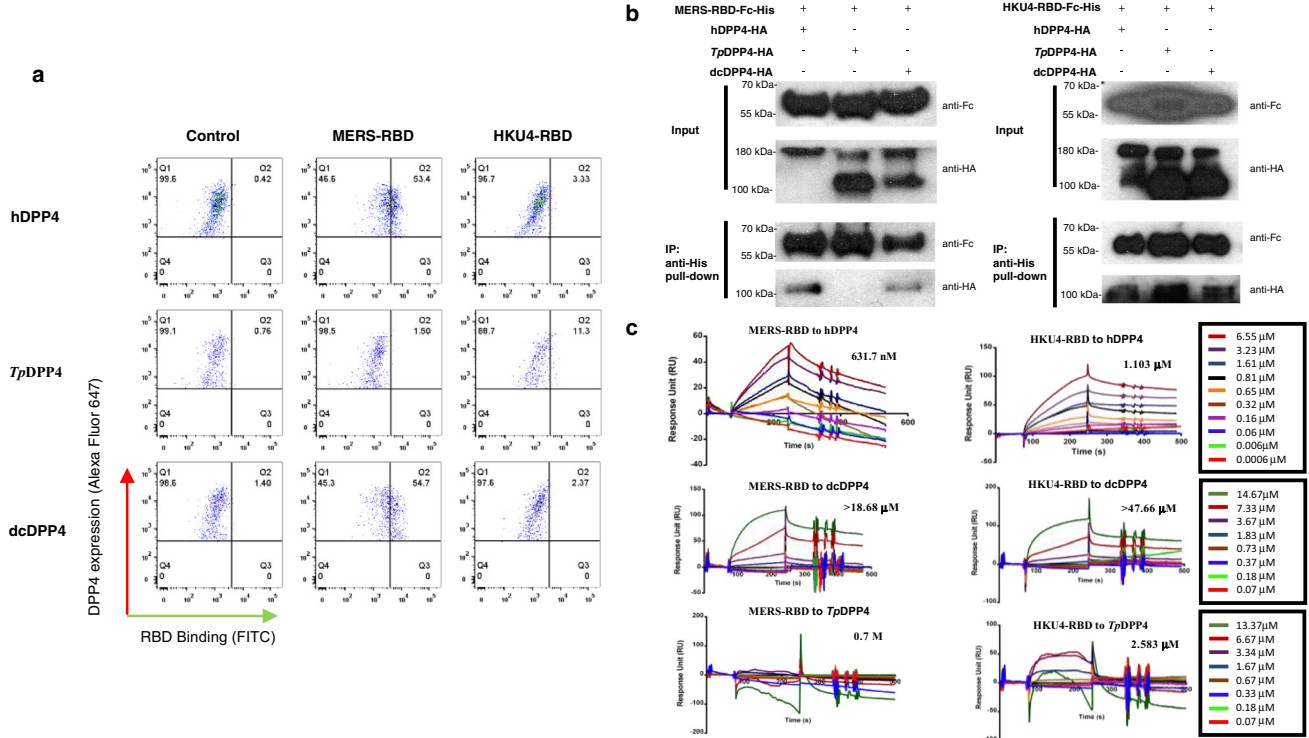

**Fig. 3 Specific interaction of MERS-RBD, HKU4-RBD with hDPP4, *Tp*DPP4, and dcDPP4. a** Characterization of binding affinity of MERS-RBD, HKU4-RBD with hDPP4, *Tp*DPP4, and dcDPP4 by flow cytometry. HEK293T cells expressing various DPP4s were detected by the Alexa Fluor 647 channel. MERS-RBD, HKU4-RBD were detected by FITC channel. **b** Characterizing interaction of MERS-RBD-Fc-His, HKU4-RBD-Fc-His with hDPP4, *Tp*DPP4, dcDPP4 by co-IP assays. DPP4s were expressed as monomer or homodimer. Images are representative of three independent experiments. Uncropped blots in Source Data are provided as Source Data file. **c** Interaction between various combinations of RBD and DPP4 characterized by SPR assay with Biacore X100. The binding profiles were shown in the form of sensorgram with gradient concentrations of DPP4 proteins.

inflammation secondary to viral infections, being most severe on 4 dpi. Histology of brain tissues on 14 dpi also showed marked perivascular lymphocytic infiltration, as well as increased lymphocytes in the brain parenchyma (Fig. 4e). Immunohistochemical (IHC) staining with mouse monoclonal antibody against *Ty*-BatCoV HKU4 S1 protein revealed positive viral antigen expression in lung alveolar septal cells and brain lymphocytes and glial cells of mice sacrificed on 4 and 14 dpi, respectively (Fig. 4f). Anti-*Ty*-BatCoV HKU4 antibody was detected in mice sacrificed on 14 dpi by western blot and IF assays (Fig. 4c, d).

**Cytokine response in *Ty*-BatCoV HKU4-infected mice.** MERS-CoV causes delayed proinflammatory response and evades innate immunity with attenuated interferon-β response, contributing to virulence[17]. To study the cytokine response during *Ty*-BatCoV HKU4 infection, the mRNA expression levels of antiviral cytokines (IFN-β, IFN-γ, and Mx1), proinflammatory cytokines (IL-1β, IL-2, IL-6, IL-12p40, and TNF-α), and chemokines (IP-10, MCP-1, MIP-1α, RANTES, CXCL-1, and G-CSF) in mouse tissues were measured using RT-qPCR assays (Supplementary Fig. 4). Antiviral cytokines were suppressed in both lung and brain tissues during *Ty*-BatCoV HKU4 infection. In lung tissues, IL-1β showed gradual increase from 4 to 14 dpi, whereas RANTES and CXCL-1 showed rapid increase on 4 dpi but decreased afterwards. In brain tissues, IL-6 showed rapid increase on 4 dpi but decreased afterwards, whereas IP-10, MIP-1α, and RANTES showed marked increase on 14 dpi.

**Discussion**
This is the first isolation of a bat CoV capable of using hDPP4 as the receptor for cell entry. Our findings carry significant public

health implications, by showing that bat CoVs closely related to MERS-CoV can potentially infect human cells and utilize the ubiquitous cellular protein for cell entry. It is considered that the RBD-receptor-binding interphase is a critical barrier for CoV cross-species transmission from bats to humans. The fact that bat CoVs can readily infect human cells suggests that they have the potential to emerge in humans by direct bat-to-human transmission without intermediate hosts. SARS-CoV was likely a recombinant virus originated from horseshoe bats, while civet was the intermediate host for bat-to-human transmission[18,19]. Although bat CoVs are notoriously difficult to cultivate, a few SARSr-BatCoV strains have been shown to replicate in Vero cells and able to utilize human angiotensin converting enzyme II (ACE2) as receptor[20,21], suggesting the potential for SARSr-BatCoVs to infect humans directly. SARS-CoV-2 was also believed to be originated from related viruses in bats, with its genome backbone closest to bat viruses and the RBD region closest to pangolin viruses[22,23]. Our results suggest that diverse bat coronaviruses not limited to SARSr-CoVs may be of potential imminent risk to human health.

Despite the considerable genetic distance from MERS-CoV (54.5% aa identities in the RBD), *Ty*-BatCoV HKU4 was able to utilize the same receptor as MERS-CoV for infecting human cells and hDPP4 transgenic mice. Intranasal inoculation of *Ty*-BatCoV HKU4 in hDPP4 transgenic mice caused a self-limiting infection, with the highest viral loads detected in the lungs on day 4 and brain on day 14, and lymphocytic inflammation observed in infected tissues. *Ty*-BatCoV HKU4 infection was associated with suppression of antiviral cytokines and activation of different proinflammatory cytokines and chemokines in lung and brain tissues, which mimics MERS-CoV infection in vitro[17]. Nevertheless, the lack of replication of *Ty*-BatCoV HKU4 in airway

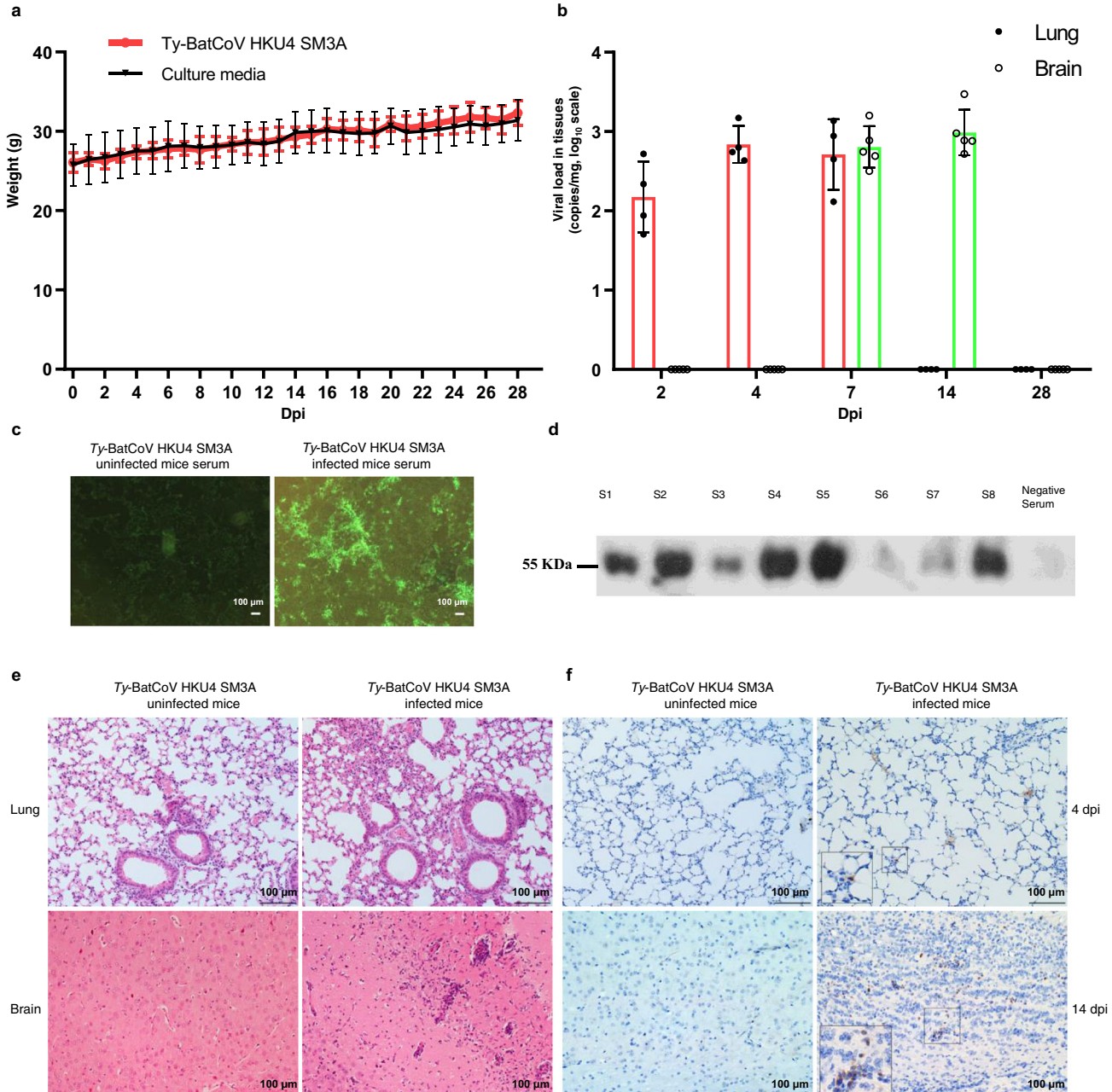

**Fig. 4 Transgenic mice with hDPP4 expression are permissive to *Ty*-BatCoV HKU4 infection. a** Weight of transgenic mice after challenge with $1 \times 10^6$ TCID$_{50}$ *Ty*-BatCoV HKU4 SM3A ($n = 3$) or culture media ($n = 3$). **b** RNA levels of *Ty*-BatCoV HKU4 SM3A detected in lung and brain tissues at 2, 4, 7, 14, and 28 dpi. Data are presented as mean values ± SD, $n = 4$ (for lung tissues) or 5 (for brain tissues) independent biological replicates at each time point. **c** IF assay showing antibodies against *Ty*-BatCoV HKU4 in virus-infected mice serum. Scale bars, 100 μm. **d** Western blot assay showing antibodies against viral N antigen in virus-infected mice serum. Representative H&E (**e**) and IHC (**f**) stained tissue sections from lungs at 4 dpi and brains at 14 dpi. Scale bars, 100 μm. Images are representative of three independent experiments. Uncropped blots in Source Data are provided as Source Data file.

epithelial cultures may suggest limited adaptation to the human respiratory tract. This is in contrast to MERS-CoV, which can infect both upper and lower airway cells[24]. Moreover, we were unable to passage *Ty*-BatCoV HKU4 in transgenic mice (unpublished data), suggesting inefficient transmission of *Ty*-BatCoV HKU4 in humans unless the virus is able to evolve rapidly upon human infection. *Ty*-BatCoV HKU4 showed purifying selection in its S protein and very few mutations were noted during passage in Caco-2 cells. Yet, the potential of *Ty*-BatCoV HKU4 and related viruses to emerge and adapt rapidly in an intermediate host or humans like SARSr-CoVs should be closely monitored. Given the genetic diversity among members of

*Merbecovirus* and the lack of serological cross-reactivity between MERS-CoV and *Ty*-BatCoV HKU4, humans with past MERS-CoV infections are unlikely protected against other merbecovirus infections. And should these viruses emerge, interferon-based treatment may be evaluated as for MERS-CoV infections.

The ability of *Ty*-BatCoV HKU4 to infect human cells and hDPP4 transgenic mice further supports a possible bat origin of MERS-CoV. However, the molecular mechanism of interspecies transmission along the evolutionary path and the role of host DPP4 remain to be elucidated. In this study, *Ty*-BatCoV HKU4 could only replicate efficiently in Caco-2 and Huh7 cells and showed limited replication in a dromedary cell line and a mouse

cell line. Although the critical residues in the DPP4 sequences were mostly conserved between human, dromedaries and lesser bamboo bats, differential binding affinities of HKU4-RBD and MERS-RBD to hDPP4, dcDPP4 and *Tp*DPP4 were observed. HKU4-RBD protein was able to bind all three proteins but with lower binding affinity to hDPP4 and dcDPP4 than MERS-RBD by SPR assays. The lower binding affinity of HKU4-RBD to *Tp*DPP4 than to hDPP4 was in-line with the inability of *Ty*-BatCoV HKU4 to replicate in primary cells derived from *Tylonycteris pachypus* and *Tp*DPP4-expressing *Tylonycteris pachypus* or HEK293T cells. This suggests that DPP4 may not be the receptor for *Ty*-Bat-CoV-HKU4 in its natural host. Although the bat ancestral origin of MERS-CoV is yet to be identified, our results suggest that MERS-CoV may have evolved from a closely related bat CoV, which could utilize dcDPP4 and hDPP4 as receptors for infections in camels and humans respectively. Besides MERS-CoV, SARS-CoV, and SARS-CoV-2, human coronavirus-229E and human coronavirus-NL63 were also likely originated from bats, which may have caused unrecognized epidemics in human history[5,25,26]. The public should be aware of the potential of bat coronaviruses to emerge and cause human epidemics, and should be refrained from disturbing the ecology of bats and consuming these important wild animals.

## Methods

**Ethics statement**. Bat samples were collected and approved by Department of Agriculture, Fisheries and Conservation, HKSAR and animals were handled in accordance with guidelines of Regulations for the Administration of Laboratory Animals (Decree No. 2 of the State Science and Technology Commission of the People's Republic of China on 14 November 1988) using standard procedures under a license from the Guangdong Entomological Institute Administrative Panel on Laboratory Animal Care (GDEI-AE-2006001); and the Committee on the Use of Live Animals in Teaching and Research at the University of Hong Kong.

**Bat sample collection**. Bats were captured from various locations in Hong Kong and mainland China over a 6-year period (August 2010 to August 2016). Alimentary samples were collected using procedures described previously[27]. All specimens were immediately placed in viral transport medium before transportation to the laboratory for RNA extraction. The bat species identity of samples positive for *Ty*-BatCoV HKU4 was confirmed by cytochrome b sequence analysis as described previously[28].

**Detection of *Ty*-BatCoV HKU4 from bat samples**. To detect *Ty*-BatCoV HKU4, viral RNA was extracted from the respiratory and alimentary samples of bats using QIAamp Viral RNA Mini Kit (Qiagen). The RNA was used as the template for RT-PCR. CoV detection was performed by amplifying a 440-bp fragment of the RdRp gene of CoVs as described previously[29]. The sequences of the PCR products were compared with known sequences of the RdRp genes of CoVs in the GenBank database.

**Spike and genome sequence analysis of *Ty*-BatCoV HKU4 strains**. To study the genetic diversity of S genes of *Ty*-BatCoV HKU4, the complete S genes of 35 *Ty*-BatCoV HKU4 strains detected were amplified and sequenced using the primers as shown in Supplementary Table 5. To study the genetic changes of *Ty*-BatCoV HKU4 during passage in infected cells, the genomes of progeny viruses of *Ty*-BatCoV HKU4 SM3A at the 1st, 5th, and 15th passages in Caco-2 cells were sequenced using the primers as shown in Supplementary Table 6. Multiple sequence alignments were constructed using Clustal W in BioEdit version 7.2.5[30,31]. Phylogenetic trees were constructed using the maximum-likelihood method[32], with bootstrap values calculated from 1000 trees. The number of synonymous substitutions per synonymous site, $K$s, and the number of non-synonymous substitutions per nonsynonymous site, $K$a, for each coding region were calculated using the Nei-Gojobori method (Jukes-Cantor) in MEGA version 5[31].

**Cell lines and isolation of *Ty*-BatCoV HKU4**. The cell lines used in this study are described in Supplementary Table 2. Primary bat cell lines were developed as described previously[33]. Original alimentary samples from lesser bamboo bats tested positive for *Ty*-BatCoV HKU4 by RT-PCR were subject to virus isolation in Caco-2 cells as described previously[5]. Cell lines were prepared in 24-well culture plates and inoculated with 300 μl of fecal samples diluted at 1:5. Nonattached viruses were removed by washing the cells twice in phosphate-buffered saline (PBS). The monolayer cells were maintained in MEM supplemented with 1% FBS. All infected

cell lines were incubated at 37 °C for 7 days. CPE were examined on day 1, 3, 5, and 7 by inverted light microscopy.

To study the cell tropism of *Ty*-BatCoV HKU4, *Ty*-BatCoV HKU4 SM3A isolated from Caco-2 cells was used to infect various cell lines including bat and camel cells, Caki-3R, Dubca, Calu-3, HeLa, HRT-18G, HT-29, Huh7, His-1, NHBE, A549, Hep-2, HFL, NT2/D1, RD, THP-1, Raji, 293 T, CaCO2, CRFK, PK15, 3T3, RK3E, LLC-MK2, Vero, Vero E6, and BS-C-1 as described previously[34]. Viral titers were determined as median tissue culture infective dose (TCID$_{50}$) per ml in confluent cells in 24-well tissue culture plates. Cells were seeded onto 24-well plates, at $2 \times 10^5$ cells per well with the respective medium and incubated at 37 °C and 5% $CO_2$ for 24 h prior to experiment. Cells were washed once with PBS and inoculated with 1 MOI *Ty*-BatCoV HKU4 SM3A for 1 h. After 1 h of viral adsorption, the supernatant was removed, and cells were washed twice with PBS.

**Antigen detection by IF**. Antigen detection of infected Caco-2 cell lines was performed by IF according to protocols described previously[35]. Cell smears at day 7 prepared and fixed in chilled acetone at −20 °C for 10 min were tested by mouse serum against *Ty*-BatCoV HKU4 nucleocapsid (N) protein. The percentages of positive cells were recorded. Uninoculated cell smear was used as negative control.

**Electron microscopy**. Negative contrast electron microscopy was performed as described previously[36,37]. Tissue culture cell extracts infected with *Ty*-BatCoV HKU4 SM3A were centrifuged at $19,000 \times g$ at 4 °C, after which the pellet was resuspended in PBS and stained with 2% phosphotungstic acid. Samples were examined with Philips CM100 transmission electron microscope.

**Viral replication studies**. To study viral replication, progeny viruses from cell culture supernatant collected at 5 dpi were subject to RT-qPCR as described previously[33]. Total RNA extracted from cell culture supernatants was reverse transcribed and amplified with *Ty*-BatCoV HKU4 primers (forward primer 5'-CGGAAAATCAACACCGGTAATGGT-3'; reverse primer 5'-TAGCC TCTGGTCCAGTCCCA-3') using real-time one-step RT-qPCR assay. Probes for *Ty*-BatCoV HKU4 [5'- (FAM)TTAAACAATTGGCYCCCAGATGGTTCTTCTA CTACA(BHQ1)-3'] were used. All experiments were performed in triplicates.

**IFN susceptibility of *Ty*-BatCoV HKU4**. Confluent Huh7 cells were pretreated overnight with IFN-α A/D or IFN-β (100 units) 24 h prior to infection. Pretreated cells were washed once with PBS, followed by inoculation with MERS-CoV EMC and *Ty*-BatCoV HKU4 SM3A at MOI 1, respectively. After 1 h of virus adsorption, the inocula were removed and the cells were washed with PBS twice. MEM supplemented with 1% FBS were added to the cells before further incubation for 72 h. Apically released progeny virus genomes were determined from supernatants collected at 0, 24, 48, and 72 h of post-infection (hpi) using RT-qPCR mentioned above.

**Neutralization assays against *Ty*-BatCoV HKU4**. Dromedary sera with neutralizing antibodies to MERS-CoV detected in previous studies were selected for neutralization assays against *Ty*-BatCoV HKU4 as described previously[3,14]. The sera were heat inactivated for 30 minutes and were serial diluted from 1:10 to 1:320. One hundred TCID$_{50}$ of *Ty*-BatCoV HKU4 SM3A was mixed and incubated for 2 h at 37 °C. The mixture was inoculated in duplicate to 96-well plates of Caco-2 cells. Results were recorded after 6 days of incubation at 37 °C.

**siRNA hDPP4-knockdown Huh7 cells and infectivity assays**. Silencer Select siRNA targeting hDPP4 proteins and negative control siRNA were transfected to Huh7 using Lipofectamine RNAiMAX transfection reagent (Invitrogen) according to the manufacturer's instructions. Briefly, confluent Huh7 cells were transfected with 10 *p*mol siRNA in serum-free medium for 6 h and siRNA-lipofectamine complex was removed after incubation. After 24 h of transfection, cells were inoculated with MERS-CoV and *Ty*-BatCoV HKU4 at MOI 1, respectively. hDPP4 mRNA expression level in cell lysates and viral load in supernatants were determined using RT-qPCR.

**CRISPR-Cas9 hDPP4-knockout Huh7 cells and infectivity assays**. LentiCRISPR-DPP4 plasmid was constructed by the subcloning of DPP4 gRNA sequence (5- CTTAGAATACAACTACGTGA-3) into the LentiCRISPR (Addgene 52961). Lentivirus expressing DPP4 gRNA and Cas9 was produced by the transfection of LentiCRISPR-DPP4, psPAX2 (Addgene 12260) and pCMV-VSV-G (Addgene 8454) into 293FT cells (Life Technologies). Huh7 cells were transduced with LentiCRISPR-DPP4 virus and further selected with puromycin dihydrochloride (Life Technologies) at 3.5 μg/ml in 10% FBS MEM. Cells were observed every 24 h for cell death. The transduced cells were subcultured to T75 culture flask for expansion when all the control cells were dead while the transduced cells stayed alive. hDPP4 expression in selected knockout cells was determined by western blot and IF staining. Purified CRISPR-Cas9 hDPP4-knockout Huh7 cells were subject to infection with *Ty*-BatCoV HKU4 SM3A. Infectivity assay was performed as previously described at MOI 1[33]. Both supernatants and cell lysates were collected

at 0, 1, 2, 3, 5 and 7 dpi. Viral load in supernatants was determined using RT-qPCR.

**Amplification and sequencing of camel and bat DPP4 mRNA transcripts.** Total RNA was extracted from dromedary camel and lesser bamboo bat cell lysates using RNeasy Mini Spin Column (Qiagen). RNA was eluted with 50 μl of RNase-free water and 1 μg of RNA was used as template for reverse transcription. The cDNA generation was performed using random hexamers and Superscript III kit (Invitrogen). Subsequently, dcDPP4 gene was amplified with following primers: forward (5′-ATG-AAGACACCGTGGAAGGTGCTCCTGGGACTGCTGGGGA-3′) and reverse (5′-CTAAGGTAGAGAGAAGCATTGCTTTAGGAAGTGGCTCATG-3′). The *Tp*DPP4 gene was amplified with following primes: forward (5′-ATGAAGAC ACCGTGGAAGGTGCTGCTGGGACTGCTGGGG-3′) and reverse (5′-CTAAG GTAAAGAGAA-GCATTGCTTTAGGAAGTGGGTCATGTG-3′). RT-PCR products were gel purified using QIAquick gel extraction kit (Qiagen) and sequenced to obtain the dromedary camel and *Tylonycteris pachypus* DPP4 mRNA sequences.

**Transfection of bat, camel, and human-DPP4 gene in cells and infectivity assays.** HEK293T cells were transiently transfected with plasmid containing *Tp*-, dc-, and hDPP4 gene with GFP gene and empty plasmid (as mock-transfected control) by Lipofectamine 2000 according to manufacturer's protocol 24 h prior to infectivity assay. Transfection of cells was validated by the expression of GFP under fluorescent microscope. The transfected cells were infected by MERS-CoV and *Ty*-BatCoV HKU4 at MOI 1 and maintained in MEM supplemented with 1% FBS before further incubation for 24 h.

**Expression and purification of S1-RBD and DPP4 proteins.** hDPP4 gene was PCR amplified from previously constructed pCAGEN-hDPP4-2A-EGFP vector[10]. Subsequently, hDPP4, dcDPP4, and *Tp*DPP4 plus HA tag were PCR amplified and cloned into pCAGEN vector using the primers as shown in Supplementary Table 7. The MERS-RBD (residues 367–606) and human codon optimized HKU4-RBD (residues 372–611) were PCR amplified and cloned into pCMV-Fc-His vector using the primers as shown in Supplementary Table 7. To prepare proteins of MERS-RBD and HKU4-RBD, the resulting plasmids pCMV-MERS-RBD-Fc-His and pCMV-HKU4-RBD-Fc-His were transfected into HEK293T cells by Lipofectamine 2000 (Invitrogen), at 1 μg DNA per $10^6$ cells. After 48 h transfection, cells were washed with PBS. Then, cells were incubated with ice-cold lysis buffer (50 mM NaH2PO4, 300 mM NaCl, 10 mM imidazole, PH 8.0) for 10 min. Lysates were transferred into 1.5 ml screw tubes for centrifugation at 4 °C, 12,000 rpm for 10 min. Supernatants were collected and then proteins were purified by Protein A/G Chromatography Cartridges (Thermo Scientific) according to the manufacturer's protocol.

**Protein binding with flow cytometry and fluorescence-activated cell sorter (FACS) analysis.** HEK293T cells were transfected with hDPP4, dcDPP4 and *Tp*DPP4 constructs respectively using Lipofectamine 2000 (Invitrogen), at 1 μg DNA per $10^6$ cells in six-well plates. After 48 h, cells were washed with ice-cold PBS and trypsinized with 10 mM EDTA in PBS. Then, cells were washed and suspended in cold PBS-10% FBS with 1% sodium azide. Next, MERS-RBD and HKU4-RBD were incubated with hDPP4, dcDPP4 or *Tp*DPP4-transfected cells, respectively, followed by incubation with 1:10 dilution of FITC-conjugated anti-His monoclonal antibody (Invitrogen) at 4 °C for 30 min. After washing with PBS-1% BSA, cells were fixed and then stained with 1:500 dilution of Alexa Fluor 647 conjugated anti-HA tag monoclonal antibody (Biolegend) at 4 °C for 30 min. The mixture was measured by flow cytometer using BD FACSDiva software according to the manufacturer's instruction with gating strategy shown in Supplementary Table 8 and Supplementary Fig. 5. FACS analysis was carried out by Flowjo software (version 10).

**Co-IP assays.** To determine the interaction between DPP4 and RBD, 1 μg of mouse anti-His monoclonal antibody (Invitrogen) and 5 μg of purified MERS-RBD-Fc-His or HKU4-RBD-Fc-His were mixed overnight at 4 °C with HEK293T cell lysates after expression of hDPP4-HA, *Tp*DPP4-HA, and dcDPP4-HA for 48 h, respectively. The formed protein complex was precipitated by 25 μl ($1 \times 10^7$) Dynabeads goat anti-mouse IgG (Invitrogen). The bounding interaction of DPP4 and RBD were analyzed by western blot. MERS-RBD-Fc-His and HKU4-RBD-Fc-His were detected by using 1:6000 dilution of goat anti-human IgG Fc (HRP) antibody (Abcam) whereas hDPP4-HA, TpDPP4-HA, and dcDPP4-HA were detected by using 1:1000 dilution of primary mouse anti-HA monoclonal antibody [HA.C5] (Abcam) and 1: 4000 dilution of secondary goat anti-mouse HRP antibody (Invitrogen).

**Bac-to-Bac baculovirus expression of bat DPP4 protein in insect cells.** The MERS-RBD and HKU4-RBD fused to a C-terminal His-tag was expressed in Sf9 insect cells grown in Sf-900 II SFM serum-free medium (Invitrogen) at 27 °C. For protein purification, Sf9 cells and supernatants were harvested 96 hpi by centrifugation and concentration, sonicated, filtrated, and loaded onto a metal-chelating resin NiNTA Agarose (Qiagen). The peptidase domain of hDPP4

(residues 39–766), dcDPP4 (residues 39–765) and *Tp*DPP4 (residues 37–764), fused to a C-terminal His-tag was expressed and purified using the same protocol as above.

**Binding assays with SPR analysis.** SPR analysis was carried out at 25 °C using BIAcore X100 machine with CM5 chips (GE Healthcare). All proteins used in the experiment were exchanged to HBS-P + buffer consisting of 0.1 M HEPS, pH7.4, 1.5 M NaCl, and 0.5% v/v Surfactant P20. RBD proteins were immobilized on the chip at a range of 3193–4370 response units (RUs). Gradient concentrations of *Tp*-/dc-/hDPP4 were then injected. After each cycle, the sensor surface was regenerated using 10 nM glycine-HCl pH 3.0. Measurements from the reference flow cell (immobilized with BSA) were subtracted from experimental values. The binding kinetics was analyzed with the software BIAevaluation Version 4.1 using 1:1 Langumuir binding and/or steading state affinity models.

**hDPP4 transgenic mice challenge.** hDPP4 transgenic mice were developed from a transgenic mouse line generated by at least five microinjection experiments. In brief, 200 C57BL/6 N fertilized oocytes were collected and microinjected with the hDPP4 transgene to ensure successful integration into the mouse genome in each experiment. The oocytes carrying the transgene were transferred to pseudo-pregnant ICR foster mother mice to produce the founder mice. Founder mice were tested for the presence of transgene using isolated DNA from earpunch biopsy by PCR with primers targeting to the hDPP4 gene (Forward: 5′-AGTACAACTA-CAACAGCCACAACGTCTATATC-3′ and Reverse: 5′-ACTG-CCCATCAGGA-GATATTGAATAATCATTGATAG-3′) as well as the GFP expression in the mice earpunch biopsy under fluorescent microscope. The genotype-positive founder mice were backcrossed with wild-type C57BL/6 N to generate offsprings carrying hDPP4 transgene genotyped. The mice were maintained at the following housing condition (room temperature: 22 ± 2 °C; relative humidity: 55 ± 10%) on a dark: light cycle of 12:12 h of artificial light.

Virus stock used for mice challenge was obtained from the 7th passage of *Ty*-BatCoV HKU4 SM3A in Caco-2 cells. hDPP4 transgenic mice (6–8 week old) were infected intranasally with approximately 80 μl ($1 \times 10^6$ TCID$_{50}$) of virus suspensions as described previously[37]. Mice challenged with culture media (MEM with 2%FBS) from uninfected cells were included as negative controls. Mice were monitored daily for signs of disease. Mice were sacrificed at 2, 4, 7, 14, and 28 dpi, respectively. After euthanasia, necropsies of mice were performed to obtain the following tissues: intestine, spleen, kidney, liver, lung, and brain. Blood was collected for antibody tests by immunofluorescence and western blot analysis.

**Detection of *Ty*-BatCoV HKU4 antibodies in mice by western blot and immunofluorescence assays.** To detect the presence of antibodies against *Ty*-BatCoV HKU4 in infected mouse sera, 50 ng of purified recombinant N protein of *Ty*-BatCoV HKU4 was used as antigen for western blot analysis. After electroblotted onto a nitrocellulose membrane (Bio-Rad), The nitrocellulose membrane was further blocked by 3% BSA-PBS at 4 °C overnight. Then, the blot was incubated with mice serum of 14 dpi with 1:300 dilution at 4 °C overnight. Followed by incubation with 1:4000 goat anti-mouse HRP (Invitrogen) at 37 °C for 1 h, the blot was developed by WesternBright ECL HRP substrate (Advansta) as described previously[3,36].

The presence of antibodies against *Ty*-BatCoV HKU4 in infected mouse sera was further confirmed by IF assay. Serum samples of *Ty*-BatCoV HKU4-infected and -uninfected mice at 2, 4, 7, 14, and 28 dpi were initially diluted 10-fold in 3% BSA-PBS as baselines and then diluted with two-fold serial dilutions. The diluted serum was further incubated with the *Ty*-BatCoV HKU4-infected Caco-2 cells in 96-wells plate, followed with incubation of 1:200 dilution of FITC-conjugated goat anti-mouse monoclonal antibody (Invitrogen) at 37 °C for 1 h. The presence of antibodies against *Ty*-BatCoV HKU4 in infected mice were observed by fluorescence microscopy (Nikon).

**Histopathological and IHC studies.** To examine the histopathology and viral replication of *Ty*-BatCoV HKU4 SM3A in tissues of challenged mice, necropsy organs of the mice were subject to both viral RNA detection by RT-PCR and immunohistological studies as described previously[35,37]. Tissues for histological examination were fixed in 10% neutral-buffered formalin, embedded in paraffin, and stained with hematoxylin and eosin (H&E). Histopathological changes were observed using Nikon 80i microscope and imaging system. Infected cell lines and tissues from challenged mice that were tested positive for *Ty*-BatCoV HKU4 by RT-PCR were subject to viral load studies and IHC staining for viral S protein. Tissue sections were subjected to antigen retrieval with pH 6.0 antigen unmasking solution (Vector Laboratories) by pressure cooker after deparaffinization and rehydration. Followed by blocking endogenous peroxidase with 0.3% $H_2O_2$ for 30 min, the tissues sections were pretreated with streptavidin solution and biotin solution at room temperature for 15 min, respectively, to avoid high background signals from the endogenous biotin or biotin-binding proteins. To minimize nonspecific signal interference, tissue sections were further blocked by incubation with M.O.M.™ Mouse Ig blocking solution (Vector Laboratories) and 1% BSA/PBS for 1 h and 30 min at 4 °C, respectively. The sections were incubated at 4 °C overnight with 1:5000 dilution of mouse anti-*Ty*-BatCoV HKU4 S1 monoclonal

antibody (Cambridge Biologics), followed with incubation of biotinylated anti-mouse IgG (1: 1000 dilution) for 10 min (Vector Laboratories). Streptavidin/peroxidase complex reagent (Vector Laboratories) was then added and incubated at room temperature for 30 min. Sections were counterstained with hematoxylin. Cells infected or uninfected by Ty-BatCoV HKU4 were included as positive and negative controls respectively in each staining. Cells were fixed in chilled acetone at −20 °C for 10 min before incubation with antibodies for staining. Color development was performed using 3,3'-diaminobenzidine and images captured with Nikon 80i imaging system and Spot-advance computer software.

**Cytokine RT-qPCR assays**. To study virus-induced cytokine profiles, lung and brain tissues of Ty-BatCoV HKU4-infected and -uninfected mice were collected at 2, 4, 7, and 14 dpi, respectively. Tissue samples were weighed and homogenized in MEM (Gbico) by TissueRuptor (Qiagen). Total RNA was extracted from tissue lysates using RNeasy Mini Kit (Qiagen). RNA was eluted in 50 μl of RNase-free water. 1 μg of RNA was converted into cDNA using random hexamers by the SuperScript III reverse transcriptase kit (Invitrogen). RT-qPCR assays for interferons (IFN-β, IFN-γ), Mx1, interleukins (IL-1β, IL-2, IL-6, IL-12p40), TNF-α, IIP-10, MCP-1, MIP-1α, RANTES, CXCL-1, and G-CSF were performed as described previously with modifications, using the primers as shown in Supplementary Table 9 with mouse β-actin for normalization[38]. RT-qPCR was performed using FastStart Universal SYBR Green Master kit (Roche) in LightCycler 2.0 (Roche). The relative expression between Ty-BatCoV HKU4-infected and -uninfected transgenic mice for each gene was calculated by the comparative ΔΔCT method.

**Reporting summary**. Further information on research design is available in the Nature Research Reporting Summary linked to this article.

## Data availability
The nucleotide sequences of the complete S and genome sequence of Ty-BatCoV HKU4 strains have been lodged within the GenBank sequence database under accession no. MW218376-MW218394 and MW218395 respectively. Source data are provided with this paper.

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

## Acknowledgements
We thank Chung-Tong Shek and members of HKSAR Department of Agriculture, Fisheries, and Conservation (AFCD) for collection of bat specimens in Hong Kong. This study was partly supported by the theme-based research scheme (project no. T11-707/15-R) of

the University Grant Committee; Health and Medical Research Fund of the Food and Health Bureau of HKSAR; Consultancy Service for Enhancing Laboratory Surveillance of Emerging Infectious Disease for the HKSAR Department of Health and the University Development Fund of the University of Hong Kong.

## Author contributions

S.K.P.L. and P.C.Y.W. conceived and planned the study. R.Y.Y.F., L.Z., K.S.M.L., A.C.P.W., H.K.H.L., E.Y.M.W., C.S.F.L., G.C.S.L., J.F., Z.H., F.C.H.F., and R.K.H.A.Y. carried out experiments and performed data analysis. S.K.P.L., L.Z., K.S.M.L., A.C.P.W., H.K.H.L., and P.C.Y.W. coordinated data analysis and wrote the manuscript. L.Z., K.H.K., and K.Y.Y. coordinated animal sampling and transgenic mouse model development. S.K.P.L., R.Y.Y.F., L.Z., K.S.M.L., and K.Y.Y. contributed equally to this work. All authors provided critical feedback and revised the manuscript.

## Competing interests

The authors declare no competing interests.
