## [Peer Review File · Nature Communications]

REVIEWER COMMENTS

Reviewer #1 (Remarks to the Author):

While ancestral relatives of MERS-CoV, such as HKU4, HKU5 and PDF-2180, have been recovered through reverse genetics, none of these viruses has been actually isolated. Here, Lau and colleagues present the first isolation of a bat merbecovirus and characterize several phenotypes of this HKU4 isolate with regard to species' cell tropism and pathogenicity in a small animal model. While this finding is entirely relevant, especially in the age of SARS-CoV-2, unfortunately, the previous work with HKU4, recovered through reverse genetics, already demonstrated many of the points made in this manuscript. Additionally, while the initial HKU4 isolation data presented here is solid, some of the other data regarding species' tropism and even replication in the murine model, is simply unclear, with textual emphasis on only some of the differences. This manuscript would benefit from additional discussion of the data and addressing the following comments.

Major comments

1. Line 78-88 and figure 2a: 293T cells also seem to show replication in this assay, similar to Huh-7 and Caco-2 cells. How do the authors explain this replication and would this have an impact on the interpretation for findings in figure 2c?
2. Line 97-101: is this data provided somewhere?
3. Figure 3a needs better labeling on the axes to clarify what is indicated by what
4. Figure 3c: how do the authors explain the differences in binding profiles for HKU4 RBD with human and bat DPP4?
5. Line 141-166: The authors need to provide more information on this mouse model. What tissues express the transgenic DPP4 and how stable is the expression? Without corresponding data on where the DPP4 is expressed, the tissue tropism described here is not as informative as it could be.
6. Line 141-166 and Figure 2b: along the same lines as made in the previous comment, the authors show a statistically significant increase in HKU4 replication on 3T3 cells, which are derived from mice. How can the authors be sure that the virus replication they see in their transgenic mice is actually coming from the transgene expression rather than wild-type mouse DPP4. While I understand mouse DPP4 has been clearly shown to fail as a receptor for MERS-CoV and very likely will not work for HKU4, the data presented here by the authors somewhat argues otherwise.

7. Line 145 and figure 4b: do the mice ever clear virus? The titer seems to still be increasing in the brain up to day 14, how would it look past that point? The authors should discuss the rationale for ending the experiment when they did and need to discuss why this virus is persistent in these animals.

8. Line 218: the authors reference data in figure 2b, which clearly shows that a camel cell line AND a mouse cell line support HKU4 replication with equivalent statistical significance (2 **), yet only mention the camel cells in this discussion.

9. Line 220-226: The authors themselves have previously shown in other studies that trypsin treatment is needed to drive entry of other HKU4 spikes in DPP4 positive cell lines. They should mention how this might also explain their findings. In other words, is it that HKU4 really uses a different receptor in bats, or is it that the primary bat cell line used here did not express the receptor (common for primary bat lines and CoV receptors) and that the protease available in the transfected human cells was not compatible with the spike – this may be bypassed with trypsin...

Minor comments

1. Line 50: it seems a little unusual to start the introduction of a MERS paper with a line only about SARS-CoV

2. Line 70: Authors obtained 19 isolates but only sequenced one of them?

3. Because the differences in 2a are fairly small, the authors might consider breaking the scale on the left axis to better elucidate what they are trying to show

4. Line 144: spell out RT as well

5. In general, the figures should be more clearly referenced in the text – calling on specific panels, for example

6. Lines 176-190: While the natural history of SARS-CoV-2 is, of course, relevant to this work, the authors should really keep the focus of this paper on MERS-CoV. This entire section drifts into discussion about (1) the receptor for SARS, (2) viral recombination and (3) pangolins. None of which are the focus of the research presented here. This section could be noticeably reduced to maintain narrative focus.

Reviewer #2 (Remarks to the Author):

The manuscript by Lau et al. reports the culmination of a substantial amount of work that led to the discovery and, importantly, isolation of a novel merbecovirus from insectivorous lesser bamboo bats found on Hong Kong island and nearby mainland China. Their work has shown the virus uses DPP4 as a cellular entry receptor. They have also demonstrated, unsurprisingly, the virus binds to DPP4 of the lesser bamboo bat with high affinity than that of human or dromedary camel DPP4. The virus also infects mice transgenic for human DPP4 with no conspicuous signs of disease and only modest immune reactivity. Overall, this is an important body of work.

A few comments for the authors to address.

L81-82. What is copies/ml? Is this viral RNA by qPCR? If so, does the assay measure only genome copies, or does it also amplify mRNA? Presumably, this RT-qPCR assay was also used for the subsequent experiments, thus it is important to know if the assay is specific for genomic vRNA. The assay amplifies from the nucleocapsid gene, which has abundant mRNA produced from it.

L184-186. Have the authors considered the paper by Boni et al. (PMID: 32724171) suggesting the virus is not a recombinant?

L497. The authors should note the construction of the plasmids. Did the receptor plasmids also contain a GFP gene? Or was the GFP plasmid co-transfected with the receptor plasmids?

Figures 2, 3, extended 1 (c and d), 3 and 4 are very difficult to read. Larger fonts would resolve this problem.

Minor comments for the authors to consider.

L26. Change to “severe acute respiratory syndrome coronavirus”

L27. Change to “Middle East respiratory syndrome coronavirus”

L50. Change to “are major reservoirs”

L53. Change to “to have originated”

L80. It may be better to state “were tested for susceptibility to Ty-BatCoV”

L89. No need to capitalize “type” because it is not a proper noun.

L100. Antibody titers are typically reported as the reciprocal of the greatest dilution with an effect. Therefore, it is most appropriate to state “titers of 10 to 160”. Also on this line, it may be clearer to state “None of the samples neutralized Ty-BatCoV”.

L109. Change “utilize” to “use”.

L127, L130. Change “while” to “whereas”

L140. Change “utilizing” to “using”

L143. Change “can be” to “was” to preserve tense

L176. Change “While” to “Although”

Reviewer #1 (Remarks to the Author):

While ancestral relatives of MERS-CoV, such as HKU4, HKU5 and PDF-2180, have been recovered through reverse genetics, none of these viruses has been actually isolated. Here, Lau and colleagues present the first isolation of a bat merbecovirus and characterize several phenotypes of this HKU4 isolate with regard to species' cell tropism and pathogenicity in a small animal model. While this finding is entirely relevant, especially in the age of SARS-CoV-2, unfortunately, the previous work with HKU4, recovered through reverse genetics, already demonstrated many of the points made in this manuscript. Additionally, while the initial HKU4 isolation data presented here is solid, some of the other data regarding species' tropism and even replication in the murine model, is simply unclear, with textual emphasis on only some of the differences. This manuscript would benefit from additional discussion of the data and addressing the following comments.

Major comments

1. Line 78-88 and figure 2a: 293T cells also seem to show replication in this assay, similar to Huh-7 and Caco-2 cells. How do the authors explain this replication, and would this have an impact on the interpretation for findings in figure 2c?

Response: We thank the reviewer for the comment. The replication in 293T cells as shown on day 5 (14.46 fold change) in Figure 2a was much lower than that in Huh-7 (2833.65 fold change) and Caco-2 (2726.33 fold change) cells. Such low level of replication is likely due to low level of hDPP4 expression in 293T cells. To account for this low level of viral replication, we have included 293T transfected with the empty plasmid (pCAGEN) as the control in Figure 2d. At 24 h, the increase in viral titre in 293T cells transfected with the empty plasmid was very low (only 1.09 fold change). Such increase was not significant when compared to those overexpressed with hDPP4 and dcDPP4. Hence, this low level of replication in 293T cells does not have significant impact towards the result interpretation in Figure 2d.

2. Line 97-101: is this data provided somewhere?

Response: We thank the reviewer for the comment. The data on cross neutralization of camel MERS sera to Ty-BatCoV HKU4 was added in Extended Data Table 3.

3. Figure 3a needs better labeling on the axes to clarify what is indicated by what

Response: We thank the reviewer for the comment. The axis labels of Figure 3a are amended for better clarity.

4. Figure 3c: how do the authors explain the differences in binding profiles for HKU4 RBD with human and bat DPP4?

Response: We thank the reviewer for this comment. The ability of HKU4 RBD in binding hDPP4 was consistent with this and previous studies (Wang, Qi et al. 2014, Yang, Du et al. 2014) showing that Ty-BatCoV HKU4 could utilize hDPP4 as receptor. The lower binding affinity to TpDPP4 when compared to hDPP4 in Figure 3c is in line with the inability of HKU4 RBD to

infect primary cells derived from *Tylonycteris pachypus* and *TpDPP4*-expressing HEK293T cells. This may imply the presence of an alternative receptor for *Ty-BatCoV HKU4* in bat host. We have provided more clear explanations in the revised discussion.

5. Line 141-166: The authors need to provide more information on this mouse model. What tissues express the transgenic DPP4 and how stable is the expression? Without corresponding data on where the DPP4 is expressed, the tissue tropism described here is not as informative as it could be.

Response: We thank the reviewer for the comment. The hDPP4 transgene expression in different tissues (kidney, liver, lung, spleen, brain, intestine) of the transgenic mice used in the study were measured by quantitative PCR. All the tissues had similar expression level of the transgene (Extended Figure 5a). Since the lung and brain tissues were more susceptible to *Ty-BatCoV-HKU4*, hDPP4 expression level especially in lung and brain tissues was measured in the infected mice at 2, 7, 14, and 28 dpi respectively (Extended Figure 5b). Our results showed that the hDPP4 transgene was stably expressed in lung and brain tissues throughout the experiment. We have also included the results in the revised manuscript.

6. Line 141-166 and Figure 2b: along the same lines as made in the previous comment, the authors show a statistically significant increase in HKU4 replication on 3T3 cells, which are derived from mice. How can the authors be sure that the virus replication they see in their transgenic mice is actually coming from the transgene expression rather than wild-type mouse DPP4. While I understand mouse DPP4 has been clearly shown to fail as a receptor for MERS-CoV and very likely will not work for HKU4, the data presented here by the authors somewhat argues otherwise.

Response: We thank the reviewer for the comment. Wild type mice (n=8) were previously employed in this study as well to determine whether *Ty-BatCoV HKU4* could infect wild mice or not. Two mice were detected for *Ty-BatCoV-HKU4* in lung tissues at 2 dpi. However, none of the mice were detected for the virus in any tissues at 4, 7, 14, and 28 dpi. Thus, we believed that *Ty-BatCoV HKU4* detected in mice at 2 dpi was only the residual virus from inoculation and that the virus could not be replicated in wild type mice. Hence, we believe that the virus replication in transgenic mice is most likely due to the hDPP4 expression. The results of *Ty-BatCoV HKU4* infection on the wild type mice is summarized as below:

Dpi 2	lung	2/8
	brain	0/8
Dpi 4	lung	0/4
	brain	0/4
Dpi 7	lung	0/4
	brain	0/4
Dpi 14	lung	0/4
	brain	0/4
Dpi 28	lung	0/4
	brain	0/4

7. Line 145 and figure 4b: do the mice ever clear virus? The titer seems to still be increasing in the brain up to day 14, how would it look past that point? The authors should discuss the rationale for ending the experiment when they did and need to discuss why this virus is persistent in these animals.

Response: We thank the reviewer for the comment. In fact, we have measured the tissue viral loads in transgenic mice for up to 28 dpi and no Ty-BatCoV-HKU4 was detected in any tissues of mice scarified at 28 dpi. Hence, we believe that the mice could clear the virus within 28 days. We have added the viral load data at 28 dpi in Figure 4b to address viral clearance.

8. Line 218: the authors reference data in figure 2b, which clearly shows that a camel cell line AND a mouse cell line support HKU4 replication with equivalent statistical significance (2 **), yet only mention the camel cells in this discussion.

Response: We thank the reviewer for the reminder. We have amended the sentence and mentioned the limited replication in a mouse cell line accordingly.

9. Line 220-226: The authors themselves have previously shown in other studies that trypsin treatment is needed to drive entry of other HKU4 spikes in DPP4 positive cell lines. They should mention how this might also explain their findings. In other words, is it that HKU4 really uses a different receptor in bats, or is it that the primary bat cell line used here did not express the receptor (common for primary bat lines and CoV receptors) and that the protease available in the transfected human cells was not compatible with the spike – this may be bypassed with trypsin...

Response: We thank the reviewer for the comment. Although exogenous trypsin was previously shown to facilitate the entry of HKU4 spike-pseudovirus in 293T cells transfected with hDPP4 and bat DPP4 from *Pipistrellus pipistrellus* and *Triatoma brasiliensis* bat lung cells (1). However, *TpDPP4* from its natural bat host was not included in that study. In contrast, we have shown that live Ty-BatCoV HKU4 was unable to infect primary *Tylosycteris pachypus* cells or 293T cells with overexpressed *TpDPP4*, even with exogenous trypsin. This implies that receptor binding instead of trypsin was more likely the limiting factor in viral infectivity. We have included the additional data on primary *Tylosycteris pachypus* cells with overexpressed *TpDPP4* in Figure 2c and more detailed explanation.

1.) Yang, Y., Du, L., Liu, C., Wang, L., Ma, C., Tang, J., Baric, R. S., Jiang, S., & Li, F. (2014). Receptor usage and cell entry of bat coronavirus HKU4 provide insight into bat-to-human transmission of MERS coronavirus. *Proceedings of the National Academy of Sciences of the United States of America*, 111(34), 12516–12521. <https://doi.org/10.1073/pnas.1405889111>

2.) Lau, S., Wong, A., Luk, H., Li, K., Fung, J., He, Z., Cheng, F., Chan, T., Chu, S., Aw-Yong, K. L., Lau, T., Fung, K., & Woo, P. (2020). Differential Tropism of SARS-CoV and SARS-CoV-2 in Bat Cells. *Emerging infectious diseases*, 26(12), 10.3201/eid2612.202308. Advance online publication.

3.) Xia, S., Lan, Q., Su, S., Wang, X., Xu, W., Liu, Z., Zhu, Y., Wang, Q., Lu, L., & Jiang, S. (2020). The role of furin cleavage site in SARS-CoV-2 spike protein-mediated membrane fusion in the presence or absence of trypsin. *Signal transduction and targeted therapy*, 5(1), 92. <https://doi.org/10.1038/s41392-020-0184-0>

Minor comments

1. Line 50: it seems a little unusual to start the introduction of a MERS paper with a line only about SARS-CoV

Response: We thank the reviewer for the comment. We have amended the sentence to better introduce MERS-CoV.

2. Line 70: Authors obtained 19 isolates but only sequenced one of them?

Response: We thank the reviewer for the comment. To ensure consistency throughout the study, we have only used one isolated strain for all experiments and sequenced this strain. We did not sequence the entire genomes of other isolates because their spike genes as sequenced from the original bat samples showed very limited variations.

3. Because the differences in 2a are fairly small, the authors might consider breaking the scale on the left axis to better elucidate what they are trying to show

Response: We thank the reviewer for the comment. We have amended the left axis scale of Figure 2a for better illustration.

4. Line 144: spell out RT as well

Response: We thank the reviewer for the comment. We have amended accordingly.

5. In general, the figures should be more clearly referenced in the text – calling on specific panels, for example

Response: We thank the reviewer for the comment. We have amended accordingly so that the figures are clearly referenced in text.

6. Lines 176-190: While the natural history of SARS-CoV-2 is, of course, relevant to this work, the authors should really keep the focus of this paper on MERS-CoV. This entire section drifts into discussion about (1) the receptor for SARS, (2) viral recombination and (3) pangolins. None of which are the focus of the research presented here. This section could be noticeably reduced to maintain narrative focus.

Response: We thank the reviewer for the comment. We have considerably reduced this paragraph to focus on MERS-CoV.

Reviewer #2 (Remarks to the Author):

The manuscript by Lau et al. reports the culmination of a substantial amount of work that led to the discovery and, importantly, isolation of a novel merbecovirus from insectivorous lesser bamboo bats found on Hong Kong island and nearby mainland China. Their work has shown the virus uses DPP4 as a cellular entry receptor. They have also demonstrated, unsurprisingly, the virus binds to DPP4 of the lesser bamboo bat with high affinity than that of human or dromedary camel DPP4. The virus also infects mice transgenic for human DPP4 with no conspicuous signs of disease and only modest immune reactivity. Overall, this is an important body of work.

A few comments for the authors to address.

L81-82. What is copies/ml? Is this viral RNA by qPCR? If so, does the assay measure only genome copies, or does it also amplify mRNA? Presumably, this RT-qPCR assay was also used for the subsequent experiments, thus it is important to know if the assay is specific for genomic vRNA. The assay amplifies from the nucleocapsid gene, which has abundant mRNA produced from it.

Response: We thank the reviewer for the comment. We confirm that copies/ml refers to viral RNA by qPCR. The assay theoretically could detect both viral genomic RNA and subgenomic RNA. However, we performed the assay using only culture supernatants which should only contain progeny viruses, while subgenomic RNA is only present within infected cells.

L184-186. Have the authors considered the paper by Boni et al. (PMID: 32724171) suggesting the virus is not a recombinant?

Response: We thank the reviewer for the comment. However, as suggested by another reviewer, we have considerably shortened this paragraph and removed the discussion on possible recombination origin of SARS-CoV-2.

L497. The authors should note the construction of the plasmids. Did the receptor plasmids also contain a GFP gene? Or was the GFP plasmid co-transfected with the receptor plasmids?

Response: We thank the reviewer for the comment. We have clarified the construction of the plasmids in line 497-498. The plasmids should be carrying both the receptor and GFP gene so that the expression of different host receptors can be reflected by the expression of GFP protein observed under the fluorescent microscope.

Figures 2, 3, extended 1 (c and d), 3 and 4 are very difficult to read. Larger fonts would resolve this problem.

Response: We thank the reviewer for the comment. We have amended the figures with larger fonts accordingly.

Minor comments for the authors to consider.

L26. Change to “severe acute respiratory syndrome coronavirus”

Response: We thank the reviewer for the comment. We have amended accordingly.

L27. Change to “Middle East respiratory syndrome coronavirus”

Response: We thank the reviewer for the comment. We have amended accordingly.

L50. Change to “are major reservoirs”

Response: We thank the reviewer for the comment. We have amended accordingly.

L53. Change to “to have originated”

Response: We thank the reviewer for the comment. We have amended accordingly.

L80. It may be better to state “were tested for susceptibility to Ty-BatCoV”

Response: We thank the reviewer for the comment. We have amended accordingly.

L89. No need to capitalize “type” because it is not a proper noun.

Response: We thank the reviewer for the comment. We have amended accordingly.

L100. Antibody titers are typically reported as the reciprocal of the greatest dilution with an effect. Therefore, it is most appropriate to state “titers of 10 to 160”. Also on this line, it may be clearer to state “None of the samples neutralized Ty-BatCoV”.

Response: We thank the reviewer for the comment. We have amended accordingly.

L109. Change “utilize” to “use”.

Response: We thank the reviewer for the comment. We have amended accordingly.

L127, L130. Change “while” to “whereas”

Response: We thank the reviewer for the comment. We have amended accordingly.

L140. Change “utilizing” to “using”

Response: We thank the reviewer for the comment. We have amended accordingly.

L143. Change “can be” to “was” to preserve tense

Response: We thank the reviewer for the comment. We have amended accordingly.

L176. Change “While” to “Although”

Response: We thank the reviewer for the comment. We have amended accordingly.

REVIEWERS' COMMENTS

Reviewer #1 (Remarks to the Author):

The authors have sufficiently addressed the major concerns I raised for their initial submission. The text is clearer and the conclusions are straightforward. This is valuable research that should be accepted for publication.